# The arch support insoles show benefits to people with flatfoot on stance time, cadence, plantar pressure and contact area

**Yu-ping Huang**[1,2☯]**, Hsien-Te Peng**[2☯]**, Xin Wang**[3]**, Zong-Rong Chen**[2,4]**, Chen-Yi Song**[5]***

**1** School of Sports Science, Nantong University, Jiangsu, China, **2** Graduate Institute of Sport Coaching Science & Department of Physical Education, Chinese Culture University, Taipei, Taiwan, **3** Department of Sports Human Science, Shenyang Sport University, Shenyang, China, **4** Department of Athletic Performance, National University of Kaohsiung, Kaohsiun, Taiwan, **5** Department of Long-Term Care, National Taipei University of Nursing and Health Sciences, Taipei, Taiwan

☯ These authors contributed equally to this work.
* cysong@ntunhs.edu.tw

## Abstract

### Background

Pes planus (flatfoot) is a common deformity characterized by the midfoot arch collapses during walking. As the midfoot is responsible for shock absorption, persons with flatfoot experience increased risk of injuries such as thumb valgus, tendinitis, plantar fasciitis, metatarsal pain, knee pain, lower-back pain with prolonged uphill, downhill, and level walking, depriving them of the physical and mental health benefits of walking as an exercise.

### Methods

Fifteen female college students with flatfoot were recruited. A wireless plantar-pressure system was used to measure the stance time, cadence, plantar pressure, and contact area. Parameters were compared between wearing flat and arch-support insoles using a two-way repeated measures ANOVA with on an incline, decline, and level surface, respectively. The significance level α was set to 0.05. The effect size (ES) was calculated as a measure of the practical relevance of the significance using Cohen's *d*.

### Results

On the level surface, the stance time in the arch-support insole was significantly shorter than in the flat insole ($p<0.05$; $ES = 0.48$). The peak pressure of the big toe in the arch-support insole was significantly greater than in the flat insole on the uphill ($p<0.05$; $ES = 0.53$) and level surfaces ($p<0.05$; $ES = 0.71$). The peak pressure of the metatarsals 2–4 and the contact area of the midfoot in the arch-support insole were significantly greater than in the flat insole on all surfaces (all $p< 0.05$).

**Data Availability Statement:** All relevant data are within the manuscript.

**Funding:** The research leading to these results has received funding from the Ministry of Science and

Technology of Taiwan (Grant No. MOST 106-2410-H-034-036), Doctoral Science and Research of Nantong University (Grant No. 135419619022) and Public Welfare Research Fund of Department of Science and Technology of Liaoning (Grant No. 20170027). No authors received a salary from any of the funders. All the funders had no role in study design, data collection and analysis, decision to publish, or preparation of the manuscript.

**Competing interests:** The authors have declared that no competing interests exist.

## Conclusions

These results imply that wearing an arch-support insole provides benefits in the shortened stance time and generation of propulsion force to the big toe while walking on uphill and level surfaces and to the metatarsals 2–4 while walking on the level surface. More evenly distributed contact areas across the midfoot may help absorb shock during uphill, downhill and level walking.

## Introduction

The foot is an integral component of the human skeletal system and plays an important role in walking. Pes planus (flatfoot) is a very common symptom frequently encountered among many diseases associated with the foot. Previous study has estimated the prevalence of mild and severe cases of the flatfoot to be 16.2% among males and 11.7% among females which was close to each other in gender [1]. Nevertheless, females are more likely to suffer from risks of lower extremity injuries in running [2] and lack of flexible foot/shank coupling coordination compared to males [3]. Approximately 1% of the population suffers from rigid flatfoot, *i.e.*, the arch height of the foot does not change between the weight-bearing and non-weight-bearing states [4, 5]. Many cases are isolated to the period while the skeletal structure is still developing: around 46% of children aged 2 to 6 years but only 14% of those 8 to 13 years suffer from the flexible flatfoot [6].

Flatfoot deformity is related to a lack of foot arch support and insufficient flexibility of the plantar ligaments and tendons [5, 7], and the collapse in the medial arch of the foot [8]. It reduces the ability to absorb the impact on the foot while walking or running [8] and can further increase the risk of foot injury and even lead to thumb valgus, tendinitis, plantar fasciitis, metatarsal pain, knee pain, lower-back pain because of the high impact forces [5, 9–12]. In people with flatfoot, the foot pronation is abnormal or excessive while walking [11–13], leading to increased fatigue in the lower extremities [8], which contributes to increased risk of the lower-extremity injury [14, 15]. As a result, people with flatfoot usually have troubles engaging in prolonged walking or running because of the lack of the load-bearing structure of the foot arch, shock dissipation and stable support of the lower extremity [9].

Foot orthoses can alleviate the symptoms of medial tibial stress [16] and significantly reduce the pain in the lower extremities [17]. They also mitigate the symptoms of lower-back pain [18] and plantar fasciitis [19]. Furthermore, orthopedic insoles demonstrated a 75.5% symptom improvement of excessive leg internal rotation, leg length discrepancy, patellofemoral pain syndrome, and plantar fasciitis symptoms [20]. Arch-support structures are often incorporated into foot orthoses for the flatfoot [17]. Previous studies have shown that arch-support insoles can reduce the peak vertical ground reaction force (GRF) in the heel by 6.9% of the body weight and increase the peak vertical GRF by 7% of the body weight [21]. They also lead to better medial-lateral control of the center of pressure of the foot, provide stability during walking [21], enable faster stair ascent time, and improve basic mobility, physical health, and comfort [17].

In daily life, people walk and run on surfaces with different slopes (*i.e.*, inclines and declines). However, it is still unknown whether flatfoot groups can keep the aforementioned advantages when walking on different slopes with arch support insoles, despite previous research evidence showed positive effects of the arch-support insole for the flatfoot [4, 16]. Hence, the purpose of this study was to investigate the effects of the arch-support insoles on

stance time, cadence, peak pressure, and contact area of the foot with the ground while walking uphill, downhill, and on a level surface, respectively. We hypothesized that wearing arch-support insoles would decrease stance time, cadence, peak pressure, and contact area on each slope walking compared to wearing flat insoles.

## Materials and methods

### Participants

Since females tended to suffering from lower extremity injuries [2] and lack of flexible foot/shank coupling coordination compared to males [3], the current study only recruited 15 female college students diagnosed as flatfoot. The average age was 19.7 ± 4.3 years, the average height was 160.9 ± 6.0 cm, and the average weight was 56.5 ± 6.7 kg. Among them, the foot width was 7.7 ± 0.5 cm at the widest point and 5.0 ± 0.7 cm at the narrowest point, on average. A priori sample size calculation was performed using GPower (version 3.1.9.2, Franz Faul, University of Kiel, Kiel, Germany), with a power level of 80% and an α level of 0.05 [22]. The expected effect size was calculated using means (15.47 and 6.52) and standard deviation (6.87 and 4.7) of the midfoot contact area under soft and hard insole conditions [23]. It revealed that the sample size of 15 participants would be sufficient for the analysis. The static arch index for each participant was calculated as the foot width at the narrowest point divided by that at the widest point times 100% [8, 24]. This is also called the Chippaux and Smirakarc index (CSI) developed by Chippaux and Smirakarch [25]. The average arch indices was 64 ± 9%. Participants with arch indexes larger than 45% on both feet were considered to have flatfoot [24]. All participants were in good health and had no prior injuries or surgeries on their lower extremities.

All participants gave written informed consent prior to the experiment. The institutional review board of Antai Medical Care Corporation, Antai Tian-Sheng Memorial Hospital (TSMH, approval number 16-107-B1) reviewed and approved the study protocol.

### Equipment

A wireless plantar-pressure insole system (Tekscan, Inc., Boston, MA, USA) was used to monitor the plantar pressure at a sampling rate of 100 Hz while each participant was walking. Wireless pressure insole with the F-Scan sensor (Model #3000E Tekscan, Inc., Boston, MA, USA) were placed above either the tested flat insole or arch-support insole of each foot to detect participant's plantar pressure. The insole consists of 960 individual pressure measuring sensors, which are referred to as sensing elements.

### Experimental protocols

The participants were instructed to walk on a treadmill with each of three slopes, a 9 degree incline (uphill walking), a -9 degree decline (downhill walking), and a level surface (level walking) [26, 27] at 0.75 m/s (2.7 km/h) speed for all slopes [26–28]. Uphill and downhill walking were performed on different days, since walking on uphill and downhill slopes should be performed on separate days with a lapse of 24 hours to avoid interference of concentric (uphill walking) and eccentric (downhill walking) contraction [29]. Level walking was assigned to be performed with either uphill or downhill walking on a same day. Participants performed a 3 min warm-up period on the treadmill at self-selected pace. Then, they walked on the adjusted treadmill for 30 seconds at one slope. There was a 6 min resting period between uphill/downhill and level walking trials.

**Table 1. Hardness of the insole.**

|  | **Flat insole** | **Arch-support insole** |
|---|---|---|
| Forefoot (pointer) | 34.2±0.8 | 20.6±1.1 |
| Midfoot (pointer) | 19±1.2 | 60.0±0.7 |
| Heel (pointer) | 34.8±0.8 | 20.6±1.5 |

All participants were asked to wear the same type of shoe (Arthur Ashe Int Low Python All Over, le coq sportif, France) with either a flat insole or arch-support insole (FOOTDISC, Global Action Inc., Taipei, Taiwan). The hardness values of the forefoot, midfoot, and heel parts of the flat insole and arch-support insole were measured using a hardness tester (Teclock GS-709N Type A; Teclock Co., Tokyo, Japan). The hardness tester was hold with both hands and pressed perpendicular to the plane of the forefoot, midfoot, and heel parts of the insole for five times, respectively. The value on the tester was recorded immediately each time. The averages of the hardness of each part of the insole are shown in Table 1.

## Data processing

Tekscan software (Tekscan Inc., Boston, MA., USA) was used to analyse the stance time, cadence, peak plantar pressure, and plantar contact area from the pressure data of trial. The stance time was evaluated as the elapsed time from when the foot contacted the ground to when it was lifted again. Cadence means the number of steps taken per minute. The peak pressure was calculated for the big toe (BT), metatarsals 1–5 (M1-M5), midfoot (MF), medial heel (MH) and lateral heel (LH). The contact areas of the forefoot (FF), MF, and heel (H) were calculated and normalized to the sum of the contact areas of the FF, MF, and H; all values are expressed as percentages of the whole contact area. The calculation for the contact area was referred to Cavanagh's (1987) arch index formula that was usually used to determine a flatter foot in a static standing posture [30]. Nevertheless, the current study used the formula to determine the contact area changes of the distribution of the FF, MF, and H when the foot was supported by the arch-support insole in dynamic walking movement. The aforementioned parameters were analysed for both feet, and the values from the two feet were averaged [31] to avoid discrepancies caused by the difference between the dominant and non-dominant legs.

## Statistical analysis

SPSS 18.0 for Windows (SPSS Science Inc., Chicago, IL, USA) was used for the statistical analyses. A two-way repeated measures ANOVA was performed to compare the parameters between trials with participants wearing flat versus arch-support insoles for each slope. Levene's test was used to test the homogeneity of the variances. A Kolmogorov-Smirnov test was used to evaluate the normality of the data, a Wilcoxon test was used when the data were not normally distributed. The Bonferroni's method was used in *post-hoc* tests where applicable. The significance level was set at $\alpha = 0.05$. The effect size (ES) for the difference between each pair of groups was calculated for each variable as a measure of the practical relevance of the significance using Cohen's *d*; ES values between 0.20 and 0.49 were considered small, those between 0.50 and 0.79 were considered moderate, and those 0.80 and above were considered large [32].

## Results

### Stance time, cadence and step frequency

Table 2 shows the comparison of stance time, cadence and step frequency between the arch support insole and flat insole in each slope. The interaction with marginal significance between

**Table 2. Comparison of stance time, cadence and step frequency.**

| | | Flat insole | | | Arch-support insole | | |
|---|---|---|---|---|---|---|---|
| | | **Left foot** | **Right foot** | **Average** | **Left foot** | **Right foot** | **Average** |
| *Stance time* [†] (ms) | | | | | | | |
| | Uphill | 748.0±92.1 | 734.7± 86.2 | 741.3±88.6 [a] | 740.0± 99.7 | 720.7±112.0 | 730.3±104.6 [a] |
| | Downhill | 659.3±79.1 | 646.0±87.6 | 652.7±81.9 [a, b] | 656.0±67.6 | 645.3±79.7 | 650.7±72.4 [a, b] |
| | Level [*] | 795.3±66.6 | 776.0±70.5 | 785.7±67.7 [b] | 766.0±79.9 | 735.3±80.1 | 750.7±77.9 [b] |
| *Cadence* [#] (step/min) | | | | | | | |
| | Uphill [a] | 51.2±6.0 | 51.2±5.9 | 51.2±5.9 | 50.7±6.2 | 51.3±7.3 | 51.0±6.6 |
| | Downhill [a, b] | 57.3±7.1 | 57.1±6.9 | 57.2±7.0 | 57.0±5.9 | 57.0±5.8 | 57.0±5.8 |
| | Level [b] | 47.9±4.3 | 48.2±4.1 | 48.0±4.2 | 49.4±4.9 | 50.2±4.7 | 49.8±4.7 |
| *Step frequency* [†] (step/min) | | | | | | | |
| | Uphill | - - - - - | - - - - - | 103.7±11.8 [a] | - - - - - | - - - - - | 102.5±12.4 [a] |
| | Downhill | - - - - - | - - - - - | 116.1±12.5 [a, b] | - - - - - | - - - - - | 117.3±11.8 [a, b] |
| | Level | - - - - - | - - - - - | 97.5±8.3 [b] | - - - - - | - - - - - | 100.1±10.0 [b] |

[†] interaction found between the insole and slope,

[*] significant difference found between the flat insole and arch-support insole,

[#] significant difference found among the slopes,

[a] significant difference found between the uphill and downhill,

[b] significant difference found between the downhill and level, $p < 0.05$.

insoles and slopes was found in the stance time ($p = 0.050$). Simple main effects of slopes showed that the stance time in the arch-support insole was significantly shorter than that in the flat insole ($p = 0.002$, $ES = 0.48$) on the level surface. Simple main effects of insoles showed that the stance time of the uphill and level surfaces was significantly longer than that of the downhill surface in the arch-support ($p = 0.019$, $ES = 0.89$; $p = 0.001$, $ES = 1.33$, respectively) and flat ($p = 0.009$, $ES = 1.04$; $p<0.001$, $ES = 1.77$, respectively) insole. There was no interaction between insoles and slopes in the cadence. The main effect of slopes showed that the cadence of the downhill surface was significantly shorter than the uphill and level surfaces ($p = 0.007$; $p<0.001$, respectively). There was no statistically significant difference found between the insoles ($p = 0.169$). In addition, the interaction with marginal significance between insoles and slopes was found in the step frequency ($p = 0.044$). Simple main effects of insoles showed that the step frequency of the uphill and level surfaces was significantly shorter than that of the downhill surface in the arch-support ($p< 0.001$, $ES = 1.22$; $p< 0.001$, $ES = 2.05$, respectively) and flat ($p = 0.004$, $ES = 1.02$; $p< 0.001$, $ES = 1.75$, respectively) insole. Simple main effects of slopes showed that the step frequency has no difference among uphill ($p = 0.132$, $ES = 0.10$), downhill ($p = 0.346$, $ES = 0.10$), and level ($p = 0.053$, $ES = 0.28$) surface.

## Peak pressure

Table 3 shows the comparison of peak pressure between the arch-support insole and flat insole in each part of the foot for each slope. The interaction between insoles and slopes was found in the BT ($p = 0.030$) and MH ($p = 0.007$).

**BT.** Simple main effects of slopes showed that the peak pressure of the BT in the arch-support insole was significantly greater than that in the flat insole on the uphill ($p = 0.002$, $ES = 0.53$) and level surface ($p = 0.019$, $ES = 0.71$). Simple main effects of insoles showed that the peak pressure of the BT of the uphill and downhill surfaces was significantly greater than

**Table 3. Comparison of peak pressure.**

| | | Flat insole | | | Arch-support insole | | |
|---|---|---|---|---|---|---|---|
| | | Left foot | Right foot | Average | Left foot | Right foot | Average |
| BT [†] (kpa) | | | | | | | |
| | Uphill [*] | 303.8±186.0 | 284.4±173.4 | 294.1±172.0 [a] | 458.5±317.1 | 356.9±236.1 | 407.7±250.0 [a] |
| | Downhill | 288.6±119.6 | 281.1±140.9 | 284.8±111.1 [b] | 370.9±208.5 | 274.5±135.3 | 322.7±133.9 [b] |
| | Level [*] | 198.9±109.0 | 143.5±82.7 | 171.2±74.7 [a, b] | 237.6±97.7 | 215.2±150.2 | 226.4±80.7 [a, b] |
| M1 [#] (kpa) | | | | | | | |
| | Uphill [c] | 245.6±87.1 | 246.3±112.2 | 245.9±96.9 | 244.1±116.3 | 275.3±155.0 | 259.7±119.5 |
| | Downhill | 189.7±104.6 | 213.3±129.7 | 201.5±110.0 | 228.2±149.7 | 212.3±147.2 | 220.2±138.8 |
| | Level [c] | 208.3±96.3 | 202.9±72.3 | 205.6±77.1 | 193.3±93.2 | 238.9±111.1 | 216.1±93.1 |
| M2 [*] (kpa) | | | | | | | |
| | Uphill | 313.9±221.4 | 294.1±192.9 | 304.0±196.3 | 326.3±239.6 | 294.9±188.5 | 310.6±212.2 |
| | Downhill | 245.6±168.6 | 256.8±206.6 | 251.2±182.0 | 304.1±206.3 | 276.6±207.2 | 290.4±196.4 |
| | Level | 354.1±216.9 | 310.5±187.2 | 332.3±196.5 | 382.9±276.7 | 351.6±239.1 | 367.3±250.5 |
| M3 [*, #] (kpa) | | | | | | | |
| | Uphill | 284.0±132.9 | 302.2±189.1 | 293.1±154.5 | 320.8±156.4 | 298.2±171.6 | 309.5±156.3 |
| | Downhill [b] | 211.5±120.6 | 225.5±145.4 | 218.5±128.6 | 279.8±166.3 | 256.8±165.6 | 268.3±158.7 |
| | Level [b] | 349.6±164.9 | 350.7±211.4 | 350.2±176.9 | 368.0±208.7 | 399.1±224.1 | 383.6±211.4 |
| M4 [*, #] (kpa) | | | | | | | |
| | Uphill | 192.0±86.1 | 205.9±110.6 | 199.0±85.1 | 217.4±115.8 | 197.2±84.0 | 207.3±89.7 |
| | Downhill [b] | 121.8±48.4 | 142.1±61.0 | 132.0±45.8 | 150.5±74.0 | 153.5±80.4 | 152.0±66.2 |
| | Level [b] | 223.9±88.7 | 228.7±100.6 | 226.3±75.6 | 236.5±97.6 | 275.1±141.0 | 255.8±98.0 |
| M5 [#] (kpa) | | | | | | | |
| | Uphill | 133.5±83.6 | 164.3±77.1 | 148.9±73.4 | 149.8±86.7 | 144.4±55.6 | 147.1±65.2 |
| | Downhill [b] | 95.7±38.2 | 114.2±51.4 | 105.0±39.2 | 111.1±48.0 | 114.0±55.5 | 112.6±46.7 |
| | Level [b] | 153.9±44.8 | 154.8±63.7 | 154.3±49.9 | 152.4±60.3 | 173.5±83.3 | 162.9±48.3 |
| MF (kpa) | | | | | | | |
| | Uphill | 140.6±91.5 | 140.2±49.8 | 140.4±56.5 | 148.2±98.2 | 122.9±46.2 | 135.6±64.2 |
| | Downhill | 96.1±30.6 | 113.8±48.1 | 105.0±34.4 | 120.1±38.4 | 120.8±53.6 | 120.4±40.2 |
| | Level | 119.9±56.4 | 123.3±47.8 | 121.6±36.7 | 108.3±27.4 | 122.4±49.4 | 115.3±31.8 |
| MH [†] (kpa) | | | | | | | |
| | Uphill [*] | 165.2±67.3 | 169.5±61.5 | 167.3±60.2 [c] | 147.8±74.5 | 140.9±62.3 | 144.3±64.4 [c] |
| | Downhill [*] | 232.5±156.2 | 211.8±116.5 | 222.1±134.2 | 197.1±136.1 | 138.7±98.4 | 167.9±114.2 |
| | Level [*] | 206.5±80.4 | 179.4±52.2 | 193.0±63.0 [c] | 165.7±90.0 | 161.7±71.2 | 163.7±72.6 [c] |
| LH [*, #] (kpa) | | | | | | | |
| | Uphill [c] | 131.5±63.6 | 146.1±102.0 | 138.8±71.7 | 111.7±61.8 | 112.2±52.4 | 112.0±49.1 |
| | Downhill | 202.3±130.2 | 161.5±80.7 | 181.9±103.1 | 164.5±111.4 | 152.7±96.9 | 158.6±99.0 |
| | Level [c] | 173.7±82.0 | 165.1±103.3 | 169.4±76.1 | 129.3±51.7 | 189.1±162.6 | 159.2±90.5 |

[†] interaction found between the insole and slope,

[*] significant difference found between the flat insole and arch-support insole,

[#] significant difference found among the slopes,

[a] significant difference found between the uphill and downhill,

[b] significant difference found between the downhill and level,

[c] significant difference found between the uphill and level, $p < 0.05$; BT = big toe, M1 = metatarsal 1, M2 = metatarsal 2, M3 = metatarsal 3, M4 = metatarsal 4, M5 = metatarsal 5, MF = midfoot, MH = medial heel, LH = lateral heel.

**Table 4. Comparison of contact area.**

| | | Flat Insole | | | Arch-support insole | | |
|---|---|---|---|---|---|---|---|
| | | Left foot | Right foot | Average | Left foot | Right foot | Average |
| FF * (%) | | | | | | | |
| | Uphill | 44.0±6.2 | 40.5±6.4 | 42.2±5.3 | 42.5±6.1 | 40.0±4.3 | 41.3±4.5 |
| | Downhill | 43.3±6.0 | 40.4±3.7 | 41.8±4.1 | 41.3±7.1 | 39.9±5.7 | 40.6±6.0 |
| | Level | 43.4±4.6 | 41.2±4.6 | 42.3±4.4 | 41.2±4.2 | 40.1±4.8 | 40.6±3.8 |
| MF * (%) | | | | | | | |
| | Uphill | 28.6±7.7 | 30.1±6.3 | 29.4±6.2 | 30.8±6.6 | 31.0±5.1 | 30.9±5.2 |
| | Downhill | 31.0±4.5 | 27.4±9.6 | 29.2±6.1 | 32.1±4.1 | 28.0±10.3 | 30.0±5.4 |
| | Level | 30.2±6.4 | 27.7±7.4 | 29.0±6.1 | 32.1±5.1 | 29.0±7.9 | 30.5±4.8 |
| H (%) | | | | | | | |
| | Uphill | 28.4±4.0 | 28.4±2.9 | 28.4±2.2 | 28.4±3.5 | 27.2±4.1 | 27.8±2.3 |
| | Downhill | 29.0±3.2 | 28.9±3.2 | 28.9±2.5 | 31.4±3.9 | 27.3±5.4 | 29.4±2.3 |
| | Level | 27.9±2.5 | 29.5±3.6 | 28.7±2.5 | 29.4±5.3 | 28.3±2.2 | 28.8±2.6 |

* significant difference found between the flat insole and arch-support insole, $p < 0.05$; FF = forefoot, MF = midfoot, H = heel.

that of the level surface in the arch-support ($p = 0.016$, $ES = 0.98$; $p = 0.033$, $ES = 0.87$, respectively) and flat ($p = 0.018$, $ES = 0.59$; $p = 0.007$, $ES = 1.20$, respectively) insole.

**MH.** Simple main effects of slopes showed that the peak pressure of the MH in the arch-support insole was significantly smaller than that in the flat insole on the uphill ($p = 0.003$, $ES = 0.37$), downhill ($p < 0.001$, $ES = 0.44$) and level surface ($p = 0.024$, $ES = 0.43$). Simple main effects of insoles showed that the peak pressure of the MH on the uphill surface was significantly smaller than that of the level surface in the flat insole ($p = 0.037$, $ES = 0.42$).

There was no interaction between insoles and slopes in the peak pressure of the M1, M2, M3, M4, M5, MF and LH. The main effect of insoles showed that the peak pressure of the M2 ($p = 0.036$), M3 ($p = 0.013$) and M4 ($p = 0.013$) in the arch-support insole was significantly greater than that in the flat insole, while the LH ($p = 0.013$) in the arch-support insole was significantly smaller than that in the flat insole. The main effect of slopes showed that the peak pressure of the M1 ($p = 0.028$) on the uphill surface was significantly greater than that of the level surface, while the peak pressure of the LH ($p = 0.028$) on the uphill surface was significantly smaller than that of the level surface. The main effect of slopes showed that the peak pressure of the M3 ($p = 0.006$), M4 ($p = 0.001$) and M5 ($p = 0.005$) on the level surface was significantly greater than that on the downhill surface.

## Contact area

Table 4 shows the comparison of contact areas of the FF, MF and H. There was no interaction between insoles and slopes in the FF, MF and H. The main effect of insoles showed that the contact area of the MF ($p = 0.001$) in the arch-support insole was significantly greater than that in the flat insole, while the contact area of the FF ($p = 0.001$) in the arch-support insole was significantly smaller than that in the flat insole.

## Discussion

The results of the current study showed that wearing the arch-support insole shortened the stance time compared with wearing the flat insole. Furthermore, the arch-support insole reduced the peak pressure on the medial heel when the foot contacted the surface during uphill, downhill and level walking. It also increased the peak pressure on the big toe, thus

assisting the foot propulsion during uphill and level walking. Moreover, the arch-support insole also increased the peak pressures on metatarsals 2–4 to assist the propulsion during walking. The arch-support insole increased the contact area of the midfoot compared with the flat insole, providing support to the medial arch, which is important for people with flatfoot.

The result of the current study showed that the stance time had an interaction effect across insole and slope. However, the stance time of the arch-support being shorter than the flat insole only occurred in level walking, but not in uphill and downhill walking. The change in the stance time upon wearing the arch-support insole could reflect a change from a pathological gait to a normal one with respect to multiple characteristics [33]. Longer stance time reflects lower gait speed that is predictive of increased mortality in middle-aged and elderly people in level walking [34]. Previous studies indicated that mid-soles composed of harder materials provide better foot support and, thereby, shorten the stance time in level walking [35]. Moreover, greater insole hardness has been shown to not only improve the efficiency of each stance but also enhance the physiological sensation during level walking [36]. The midfoot of the arch-support insole used in the current study was harder than that of the flat insole and, as a result, the arch-support insole was associated with a significantly shorter stance time than the flat insole during level walking on a treadmill. It was suggested that the arch-support insole provided better support for the midfoot, which effectively shortened the stance time when walking on a level treadmill and may increase the gait speed when walking on the level ground [34].

Although the peak pressure of the MH demonstrated an interaction effect across insole and slope, it was significantly reduced in the arch-support insole during uphill, downhill and level walking compared to the flat insole. Walking requires periodic motion of the lower extremities, and many injuries to the lower extremities are related to the loading impact and over-pronation of the foot [13, 37]. Previous studies have shown that most of the physical stresses associated with walking are concentrated around the medial calcaneal tubercle [38]. Furthermore, the flatoot is closely associated with pain in the medial calcaneal tubercle [39, 40]. The function of the arch-support insole is thought to help restore the elasticity of the foot arch [41] for people with flatfoot, thereby reducing the foot pronation during uphill, downhill and level walking. The reduced medial heel peak pressure in the arch-support insole was speculated to reduce the heel striking impact and further facilitate the transfer of loading to improve stability and comfort [42] that would benefit people with flatfoot.

The peak pressure of the BT has an interaction effect across insole and slope in the current study. It was significantly greater in the arch-support insole than in the flat insole during uphill and level walking. Most people used the heel-strike strategy, in which the midfoot is responsible for transferring the plantar weight-bearing from the heel to the forefoot. The forefoot plays a crucial role in propulsion in the last phase of a stride [23, 43]. The BT of the forefoot in the arch-support insole was suggested to provide propulsion in uphill and level walking. Moreover, participants exhibited greater peak pressures on metatarsals 2–4 of the forefoot while wearing the arch-support insole than with the flat insole during level walking. This effect can be explained as an optimization of the heel lift by sharing a part of the heel load and assisting the foot propulsion during walking [43]. The arch-support insole may restore the function of medial foot arch, thereby facilitating the natural elastic stretch of the plantar fascia tension associated with the windlass mechanism [38, 44].

In the current study, the proportion of the contact area of the midfoot to that of the whole foot was greater with the arch-support insole than with the flat insole during uphill, downhill, and level walking. The finding is in agreement with a previous study showing that foot orthotics with a 30˚ inverted angle can increase the contact area of the medial midfoot [45]. Using the arch-support insole like using the foot orthotics inserted beneath the midfoot makes the

contact area of the midfoot increased during walking. This increase in midfoot contact area in people with flatfoot may be associated with the support of the medial longitudinal foot-arch [46]. It was suggested that arch-support insoles can extend the contact area more evenly across in the forefoot, midfoot, and heel and, therefore, may better disperse the cumulative foot pressure over time to reduce the risk of soft-tissue injuries to the foot [47] during walking.

There were some limitations in the current study. The uphill, downhill, and level walking were performed on a treadmill in a laboratory, which did not realistically replicate the performance in an outdoor environment. Our finding did not apply to the male counterpart and people without the flatfoot problem. Only the static footprint index was used to evaluate the flatfoot in the current study, even though there are many methods, such as the navicular drop, rear foot eversion, foot posture index values, and many others, to determine the flatfoot population in the clinic [48]. The current study did not assess arch deformation in any way, thus discussion regarding the windlass mechanism were only speculations. In addition, the arch support insole did have higher hardness in the midsole and the flat insole was harder in the forefoot and the heel. This difference in stiffness could be a confounding variable to the results in the study and therefore, we cannot conclude that the differences were due to differences in hardness or the presence vs absence of an arch support.

## Conclusions

These results imply that wearing an arch-support insole provides benefits in the shortened stance time. The arch-support insole helps absorb shock at the medial heel during uphill, downhill, and level walking, provided big toe propulsion during uphill and level walking, and applied metatarsals 2–4 propulsion during level walking compared with the flat insole. Furthermore, it facilitates more evenly distributed contact area over the entire foot. Based on these findings, we recommend that people with flatfoot wear arch-support insoles to restore the function of an elastic foot arch.

## Supporting information

**S1 Data.**
(ZIP)

## Author Contributions

**Conceptualization:** Yu-ping Huang, Hsien-Te Peng.

**Data curation:** Chen-Yi Song.

**Formal analysis:** Yu-ping Huang, Hsien-Te Peng, Xin Wang, Chen-Yi Song.

**Methodology:** Yu-ping Huang, Hsien-Te Peng, Xin Wang.

**Supervision:** Chen-Yi Song.

**Writing – original draft:** Yu-ping Huang, Hsien-Te Peng, Chen-Yi Song.

**Writing – review & editing:** Zong-Rong Chen, Chen-Yi Song.

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
