## [Decision Letter · Decision Letter 0]

7 Apr 2020

PONE-D-20-05316

The arch support insoles show benefits to people with flatfoot on gait process time, plantar pressure and contact area

PLOS ONE

Dear Dr Chen-Yi Song,

Thank you for submitting your manuscript to PLOS ONE. After careful consideration, we feel that it has merit but does not fully meet PLOS ONE’s publication criteria as it currently stands. Therefore, we invite you to submit a revised version of the manuscript that addresses the points raised during the review process.

We would appreciate receiving your revised manuscript by May 22 2020 11:59PM. To enhance the reproducibility of your results, we recommend that if applicable you deposit your laboratory protocols in protocols.io, where a protocol can be assigned its own identifier (DOI) such that it can be cited independently in the future. For instructions see: http://journals.plos.org/plosone/s/submission-guidelines#loc-laboratory-protocols

We look forward to receiving your revised manuscript.

Kind regards,

Gianluca Vernillo, Ph.D.

Academic Editor

PLOS ONE

"The research leading to these results has received funding from the Ministry of Science and Technology of Taiwan (Grant No. MOST 106-2410-H-034-036), Doctoral Science and Research of Nantong University (Grant No. 135419619022) and Public Welfare Research Fund of Department of Science and Technology of Liaoning (Grant No. 20170027)."

3.  We note that the Figures in your submission might contain copyrighted images. All PLOS content is published under the Creative Commons Attribution License (CC BY 4.0), which means that the manuscript, images, and Supporting Information files will be freely available online, and any third party is permitted to access, download, copy, distribute, and use these materials in any way, even commercially, with proper attribution. For more information, see our copyright guidelines: http://journals.plos.org/plosone/s/licenses-and-copyright.

1.    You may seek permission from the original copyright holder of the Figures to publish the content specifically under the CC BY 4.0 license.

4. Please include your tables as part of your main manuscript and remove the individual files. Please note that supplementary tables (should remain/ be uploaded) as separate "supporting information" files

Reviewers' comments:

Reviewer's Responses to Questions

**Comments to the Author**

1. Is the manuscript technically sound, and do the data support the conclusions?

Reviewer #1: Partly

Reviewer #2: Partly

2. Has the statistical analysis been performed appropriately and rigorously? 

Reviewer #1: Yes

Reviewer #2: No

3. Have the authors made all data underlying the findings in their manuscript fully available?

Reviewer #1: Yes

Reviewer #2: Yes

4. Is the manuscript presented in an intelligible fashion and written in standard English?

Reviewer #1: Yes

Reviewer #2: Yes

5. Review Comments to the Author

Reviewer #1: General comments:

I would like to thank the editor for inviting me to review the manuscript entitled “The arch support insoles show benefits to people with flatfoot on gait process time, plantar pressure and contact area”. This study compared various plantar pressure measurements while walking on three slopes and while wearing two different types of insoles. Please find below my comments, which may aid the authors with submitting a revised version of their manuscript.

- I think that the authors made a great attempt to link their variables of interest to measures of importance to people with flatfeet. However, the authors attempted to link simple plantar pressure variables to mobility and walking benefits without describing what is meant by these terms. I would recommend to clearly state what is meant by the terms “mobility” and “benefits” and how improvements in mobility look like.

- One of the authors’ main conclusions, namely that increased midfoot area relative to total area in the arch-support insole is representative of the restoring of a “normal” arch is, in my opinion, an incorrect statement. If the loaded midfoot area increases, it means that more of the midfoot is in contact with the ground. This would mean that the foot is actually in a more flattened position. I would strongly recommend revising the statements made upon these findings.

- Lastly, I would recommend removing figure 1, as it does not provide valuable information to the understanding of the methods of this paper. I would also recommend providing all units within tables 1-4 instead of including them in the table headings. The units in table 1 are unclear. Also, the p-values and Cohen’s d effect sizes should be provided for all comparisons in all tables.

Specific comments:

Line 43: In the introduction of the main text, you have not mentioned any previous study investigating increased pressure at the hip, knee, or ankle joints in people with flat feet; but you talk about it in the background section of the abstract. Please either introduce the observed changes at the hip, knee, and ankle joints from the previous studies in the introduction of the main text or delete this statement from the background section of the abstract. Also, I would strongly recommend to change the wording from “pressure on the lower back, hip, …” to “loading on the lower back, hip, …”.

Line 44: “… and risk serious damage to these joints …”. You have not provided any evidence for this statement in the main text. Please either remove this statement from the abstract or elaborate on this and provide references in the main text.

Line 51: Please consider rephrasing the sentence to “The significance level α was set to 0.05.”

Lines 52-58: Please include Cohen’s d effect sizes for the reported comparisons. Also, please reorder the results section so that you always mention the same insole first in your comparisons. This would make it easier for the reader to follow your findings.

Line 60: It is unclear how your reported findings “provide benefits in mobility”. You cannot make this statement with the results that you presented in the above section unless you clearly demonstrate how these variables are related to benefits in mobility.

Line 74: “… suffers from rigid the flatfoot …”: remove “the” from the sentence

Line 80: Please include a reference at the end of the sentence.

Lines 83-84: This sentence suggests that flatfoot can result in damage to internal organs and the brain. This is incorrect. Please remove this statement from the manuscript.

Line 87: Please include a reference at the end of the sentence.

Line 91: I think using the term “cure rate” is inappropriate for two reasons: 1) leg internal rotation and leg length discrepancy is not a “disease” and therefore cannot be “cured”, 2) wearing insoles does not “cure” anything, it may reduce excessive leg internal rotation or offset leg length discrepancy. Please replace the term “cure rate” with a more appropriate term.

Line 95: the peak vertical ground reaction force is not the propulsion force. They are probably related to each other but they are not the same. Therefore, please remove “(i.e., propulsion)” from the sentence.

Lines 102-104: this sentence is the same as the sentence before that. Please remove this sentence and include its references to the previous sentence.

Lines 106-108: this is not a hypothesis. Clearly state how wearing the arch-insole will affect your variables of interest.

Line 112: remove “with” from the sentence.

Line 116: Consider rephrasing the sentence to: “A priori sample size calculation was performed using GPower (Company, City, Country), …”.

Line 118: Why were the means and standard deviations for the medial-lateral COP used to perform the sample size calculations if these are not the variables of interest in your study?

Line 122: the plural of “index” is “indices”

Line 123: “… were considered to have the flatfoot …”. Remove “the” from the sentence.

Lines 125-127: the review board that approved this study is not affiliated with any institution of any of the authors. Could you please elaborate on that?

Line 134-135: Please consider rephrasing the sentence to: “The insole consists of 960 individual pressure measuring sensors.”

Lines 142-143: Please consider rephrasing the sentence to: “Each subject performed a 3 min warm-up period on the treadmill at self-selected pace.”

Lines 143-147: In one sentence you say that participants walked on all slopes one after the other with a 6 min break in between slopes. In the next sentence, however, you state that walking on different slopes was performed on different days. Those are contradicting statements. Which one is true?

Line 151: How was the hardness measured exactly. More information is needed.

Line 158: “gait process time”. This variable is typically referred to as “stance time” or “ground contact time”. Please consider changing this throughout the manuscript to not confuse the reader.

Line 161: remove “each” from the sentence.

Line 177: Please include a reference for how Cohen’s d effect sizes were calculated.

Line 185: was there a difference in stance times for the other slopes? Even if not, please include a sentence stating so and also include the p-value and effect size for the comparison.

Lines 190-212: Please change the order of reporting your comparisons. Sometimes you first mention the arch-support insole and other times you first mention the flat insole. Please pick one insole that you will always mention first in your comparisons. This would confuse the reader less.

Lines 210-211: please include your p-values and ES right after the slope condition.

Lines 217-219: you said that the arch-support insole shortened the time of each gait cycle. This is incorrect. A gait cycle goes from the heel strike of one foot to the heel strike of the same foot. Your findings showed, however, that the stance times were altered. Please revise.

Line 218: you state that mobility was improved. None of your variables of interest is a measure for mobility. At least you did not make good enough of a case that a variable you investigated actually represents mobility. Please provide more evidence that stance times are a legitimate measure for mobility or remove this statement from the paper.

Line 220: you state that the arch-support insole absorbed the shock associated with the foot contacting the ground. Again, this is a pure speculation and you have not provided any evidence that this is the case. Please provide the evidence or delete this statement from the manuscript.

Lines 238-239: I do not understand this sentence. Please elaborate what you want to say here.

Lines 244-245: It is only a speculation that wearing arch-support insoles will “restore a normal, elastic arch in people with flatfoot, …”. Please clearly state that this is only a speculation or include a reference that has shown this in the past. Also, what is a “normal” arch?

Lines 247-249: You cannot make this statement. You have not demonstrated why reducing peak pressures would be a benefit for people with flatfoot. Please make a better case of how you link peak pressure to substantial benefits in people or remove this statement.

Lines 256-258: How does wearing an arch-support insole restore a “normal”, elastic arch? What do you speculate is the mechanism behind it? Again, what is “normal”? Perhaps, you should not use this term.

Lines 258-265: You have not assessed arch deformation in any way in this study; yet, you discuss the windlass mechanism and how the arch may or may not collapse with these insoles. I would strongly recommend highlighting that these are speculations, only, or remove the entire paragraph from the manuscript.

Lines 269: how would the forefoot play a crucial role in the braking during downhill walking if all participants were striking with the heel? The forefoot will mainly still be in the air during the braking period of the stance phase. Therefore, the forefoot cannot contribute to the braking at all. Again, this is a speculation because you have not assessed forefoot motion. Please remove this statement from the paper.

Lines 276-278: If the area of the midfoot increases relative to the total area, then the foot is flatter compared to when the midfoot area is lower compared to the total area. This is also called the arch index (Cavanagh, 1987). Your conclusion, therefore, does not make sense. Please revise.

Line 281: remove “make” from the sentence.

Lines 293-294: you cannot make the statement that wearing the arch-support insole provided many benefits in mobility. You have not provided clear reasoning as to why this would be the case. Please remove this statement from the paper.

Reviewer #2: The purpose of this study was to determine if persons with flatfoot (i.e., per planus) changed the pressure distribution across the foot while walking uphill, downhill, or on even ground in flat or arch supported insoles. The study indicated that the “gait process time”, the pressure distribution, and contact area were different in the flat or arch support insoles, but seemed to be dependent on the walking condition. I believe there needs to be some improvements in the analysis and interpretation before this manuscript is ready for publication.

I have a number of minor comments, but those may become irrelevant after the major comments are addressed. Here are my major comments.

1. The statistics need to change. There was a comparison of insole condition across different levels of incline. The results seemed to indicate that the variables that were different across insole conditions seemed to be dependent on the level of incline. With this observation in mind, the researchers should re-run the statistics on each dependent variable and instead of conducting multiple uncorrected t-tests, they should conduct a two way repeated measures ANOVA with factors INCLINE (uphill, level, downhill) and INSOLE (flat, arch support). This will indicate if there is an interaction effect for the measured pressure and gait variables.

2. Why did you restrict you sample to only females, particularly when the condition seems to be more common in males? Please provide an explanation why there wasn’t any males.

3. Please provide an explanation as to why your static arch index was representative of flatfoot. My understanding is that flat foot would have something to do with the proportion of the foot that is or is not in contact with the ground in a loaded or unloaded condition. What does the narrowest and widest part of the foot have to do with the arch collapsing or per planus?

4. Where was the pressure insole placed relative to the insoles (i.e., underneath or above)? Please provide an explanation why you used on placement method over the other.

5. How much of the results were due to hardness/stiffness of the insole versus it having arch support or not? For example, there was a difference in midsole hardness in the arch support condition and this condition also presented with an increase contact area. The same results applies to the forefoot area such that the harder condition showed increase contact area.

6. Please change “gait process time” to stance time that would be more consistent with the literature.

7. Please present the data across the dominant and non-dominant leg. If in one leg there was higher peak pressure and lower peak pressure in the other, this would average out to no change in pressure. I don’t think your averaging method avoids discrepancies.

Please incorporate these changes and adjust the discussion accordingly. Once those comments are addressed, I will be able to provide more feedback on the minor, or major, comments that remain.

6. PLOS authors have the option to publish the peer review history of their article (what does this mean?). If published, this will include your full peer review and any attached files.

Reviewer #1: Yes: Sasa Cigoja

Reviewer #2: No

---

## [Author Response · Author response to Decision Letter 0]

27 May 2020

RESPOND to Reviewer #1:

General comments: 

I would like to thank the editor for inviting me to review the manuscript entitled “The arch support insoles show benefits to people with flatfoot on gait process time, plantar pressure and contact area”. This study compared various plantar pressure measurements while walking on three slopes and while wearing two different types of insoles. Please find below my comments, which may aid the authors with submitting a revised version of their manuscript.

- I think that the authors made a great attempt to link their variables of interest to measures of importance to people with flatfeet. However, the authors attempted to link simple plantar pressure variables to mobility and walking benefits without describing what is meant by these terms. I would recommend to clearly state what is meant by the terms “mobility” and “benefits” and how improvements in mobility look like. 

RESPOND:

Thank you for your comment. We have changed the term “mobility” to gait speed. Studenski, et al. (2011) indicated that longer stance time reflects lower gait speed that is predictive of increased mortality in middle-aged and elderly people. The shortened stance time found in the current study was speculated to increase the gait speed because of wearing the arch-support insole. In addition, we have removed this statement from conclusion that “wearing the arch-support insole provided many benefits in mobility”. Please refer to lines 242-247. These results imply that wearing an arch-support insole provides benefits in the shortened stance time and generation of propulsion force to the big toe while walking on uphill and level surfaces and to the metatarsals 2-4while walking on the level surface. We have stated these in abstract and conclusion section. Please refer to lines 61-63 and 306-310.

Reference:

Studenski S, Perera S, Patel K, Rosano C, Faulkner K, Inzitari M, et al. Gait speed and survival in older adults. Jama. 2011; 305(1): 50-58. doi:10.1001/jama.2010.1923

- One of the authors’ main conclusions, namely that increased midfoot area relative to total area in the arch-support insole is representative of the restoring of a “normal” arch is, in my opinion, an incorrect statement. If the loaded midfoot area increases, it means that more of the midfoot is in contact with the ground. This would mean that the foot is actually in a more flattened position. I would strongly recommend revising the statements made upon these findings.

RESPOND:

Thank you for your comment. We have revised all the statements and removed the word “normal” throughout the manuscript.

- Lastly, I would recommend removing figure 1, as it does not provide valuable information to the understanding of the methods of this paper. I would also recommend providing all units within tables 1-4 instead of including them in the table headings. The units in table 1 are unclear. Also, the p-values and Cohen’s d effect sizes should be provided for all comparisons in all tables.

RESPOND:

Thank you for your comment. We have removed figure 1. We provide all units within tables 1-4. We changed the statistical method to a repeat-measures two-way ANOVA according to another reviewer’s comment. There would be many p-values and Cohen’s d effect sizes. It was hard to present them in tables. So they were provided in the text of the manuscript. Please refer to tables 1-4 and results section.

Specific comments:

Line 43: In the introduction of the main text, you have not mentioned any previous study investigating increased pressure at the hip, knee, or ankle joints in people with flat feet; but you talk about it in the background section of the abstract. Please either introduce the observed changes at the hip, knee, and ankle joints from the previous studies in the introduction of the main text or delete this statement from the background section of the abstract. Also, I would strongly recommend to change the wording from “pressure on the lower back, hip, …” to “loading on the lower back, hip, …”.

RESPOND:

Thank you for your comment. We have revised the abstract and mentioned that the flatfoot would increase the risk of foot injury and even lead to thumb valgus, tendinitis, plantar fasciitis, metatarsal pain, knee pain, lower-back pain conducted from previous studies. Please refer to lines 43-44.

Line 44: “… and risk serious damage to these joints …”. You have not provided any evidence for this statement in the main text. Please either remove this statement from the abstract or elaborate on this and provide references in the main text.

RESPOND:

Thank you for your comment. We have removed this statement from the abstract and mentioned that the flatfoot would increase the risk of foot injury and even lead to thumb valgus, tendinitis, plantar fasciitis, metatarsal pain, knee pain, lower-back pain conducted from previous studies. We also elaborated on this and provide references in the main text. Please refer to lines 43-44 and 81-84.

Line 51: Please consider rephrasing the sentence to “The significance level α was set to 0.05.”

RESPOND:

Thank you for your comment. We have revised the sentence. Please refer to lines 50-51.

Lines 52-58: Please include Cohen’s d effect sizes for the reported comparisons. Also, please reorder the results section so that you always mention the same insole first in your comparisons. This would make it easier for the reader to follow your findings.

RESPOND: 

Thank you for your comment. We have revised the sentence. Please refer to lines 54-59.

Line 60: It is unclear how your reported findings “provide benefits in mobility”. You cannot make this statement with the results that you presented in the above section unless you clearly demonstrate how these variables are related to benefits in mobility.

RESPOND:

Thank you for your comment. We have revised the sentence. Please refer to lines 61-63.

Line 74: “… suffers from rigid the flatfoot …”: remove “the” from the sentence

RESPOND:

Thank you for your comment. We have removed “the” from the sentence. Please refer to line 74.

Line 80: Please include a reference at the end of the sentence.

RESPOND:

Thank you for your comment. We include a reference at the end of the sentence. Please refer to line 80.

Lines 83-84: This sentence suggests that flatfoot can result in damage to internal organs and the brain. This is incorrect. Please remove this statement from the manuscript.

RESPOND:

Thank you for your comment. We have removed this statement from the manuscript. Please refer to lines 82-84.

Line 87: Please include a reference at the end of the sentence. 

RESPOND:

Thank you for your comment. We have revised the sentence and include a reference at the end of the sentence. Please refer to line 88.

Line 91: I think using the term “cure rate” is inappropriate for two reasons: 1) leg internal rotation and leg length discrepancy is not a “disease” and therefore cannot be “cured”, 2) wearing insoles does not “cure” anything, it may reduce excessive leg internal rotation or offset leg length discrepancy. Please replace the term “cure rate” with a more appropriate term.

RESPOND: 

Thank you for your comment. We have revised the sentence. Please refer to line 92.

Line 95: the peak vertical ground reaction force is not the propulsion force. They are probably related to each other but they are not the same. Therefore, please remove “(i.e., propulsion)” from the sentence.

RESPOND: 

Thank you for your comment. We have removed “(i.e., propulsion)” from the sentence. Please refer to lines 94-96.

Lines 102-104: this sentence is the same as the sentence before that. Please remove this sentence and include its references to the previous sentence.

RESPOND: 

Thank you for your comment. We have removed this sentence and include its references to the previous sentence. Please refer to lines 98, 100-101.

Lines 106-108: this is not a hypothesis. Clearly state how wearing the arch-insole will affect your variables of interest. 

RESPOND: 

Thank you for your comment. We have revised the sentence. Please refer to lines 106-108. 

Line 112: remove “with” from the sentence. 

RESPOND: 

Thank you for your comment. We have removed “with” from the sentence. Please refer to lines 112.

Line 116: Consider rephrasing the sentence to: “A priori sample size calculation was performed using GPower (Company, City, Country), …”.

RESPOND: 

Thank you for your comment. We have revised the sentence. Please refer to line 116.

Line 118: Why were the means and standard deviations for the medial-lateral COP used to perform the sample size calculations if these are not the variables of interest in your study? 

RESPOND: 

Thank you for your comment. We have changed the variable to “contact area” to perform the sample size calculations. Please refer to lines 117-119. 

Line 122: the plural of “index” is “indices”

RESPOND: 

Thank you for your comment. We have revised the word. Please refer to lines 123.

Line 123: “… were considered to have the flatfoot …”. Remove “the” from the sentence.

RESPOND: 

Thank you for your comment. We have removed the word. Please refer to lines 123-124.

Lines 125-127: the review board that approved this study is not affiliated with any institution of any of the authors. Could you please elaborate on that?

RESPOND: 

The experiment was practiced in Chinese Culture University. And all participants were recruited from Chinese Culture University. Chinese Culture University does not have its own review board. Chinese Culture University usually cooperates with Antai Tian-Sheng Memorial Hospital. That’s why the review board approved this study is not affiliated with any institution of any of the authors.

Line 134-135: Please consider rephrasing the sentence to: “The insole consists of 960 individual pressure measuring sensors.”

RESPOND: 

Thank you for your comment. We have revised the sentence. Please refer to lines 135-136.

Lines 142-143: Please consider rephrasing the sentence to: “Each subject performed a 3 min warm-up period on the treadmill at self-selected pace.”

RESPOND: 

Thank you for your comment. We have revised the sentence. Please refer to lines 145-146.

Lines 143-147: In one sentence you say that participants walked on all slopes one after the other with a 6 min break in between slopes. In the next sentence, however, you state that walking on different slopes was performed on different days. Those are contradicting statements. Which one is true?

RESPOND: 

Thank you for your comment. Uphill and downhill walking were performed on different days, since walking on uphill and downhill slopes should be performed on separate days with a lapse of 24 hours to avoid interference of concentric (uphill walking) and eccentric (downhill walking) contraction [27]. Level walking was assigned to be performed with either uphill or downhill walking on a same day. Participants performed a 3 min warm-up period on the treadmill at self-selected pace. Then, they walked on the adjusted treadmill for 30 seconds at one slope. There was a 6 min resting period between uphill/downhill and level walking trials. We have revised the sentences. Please refer to lines 141-145.

Line 151: How was the hardness measured exactly. More information is needed.

RESPOND: 

Thank you for your comment. We have increase more information. Please refer to lines 151-156.

Line 158: “gait process time”. This variable is typically referred to as “stance time” or “ground contact time”. Please consider changing this throughout the manuscript to not confuse the reader. 

RESPOND: 

Thank you for your comment. We have revised this throughout the manuscript. Please refer to lines 48, 54, 67, 107, 161, 163, 189, 190, 192, 193, 194, 250, 252, 254 and 258.

Line 161: remove “each” from the sentence.

RESPOND: 

Thank you for your comment. We have removed “each” from the sentence. Please refer to lines 161-162.

Line 177: Please include a reference for how Cohen’s d effect sizes were calculated. 

RESPOND: 

Thank you for your comment. We include a reference in the sentence. Please refer to line 170.

Line 185: was there a difference in stance times for the other slopes? Even if not, please include a sentence stating so and also include the p-value and effect size for the comparison.

RESPOND: 

Thank you for your comment. We have revised the sentence. Please refer to lines 199-200 and table 2. 

Lines 190-212: Please change the order of reporting your comparisons. Sometimes you first mention the arch-support insole and other times you first mention the flat insole. Please pick one insole that you will always mention first in your comparisons. This would confuse the reader less.

RESPOND: 

Thank you for your comment. We have revised these sentences and always keep the arch-support insole be first mentioned. Please refer to lines 205-237. 

Lines 210-211: please include your p-values and ES right after the slope condition.

RESPOND: 

Thank you for your comment. We have revised the sentence. Please refer to lines 208-218. 

Lines 217-219: you said that the arch-support insole shortened the time of each gait cycle. This is incorrect. A gait cycle goes from the heel strike of one foot to the heel strike of the same foot. Your findings showed, however, that the stance times were altered. Please revise. 

RESPOND: 

Thank you for your comment. We have revised the sentence. Please refer to lines 242-243. 

Line 218: you state that mobility was improved. None of your variables of interest is a measure for mobility. At least you did not make good enough of a case that a variable you investigated actually represents mobility. Please provide more evidence that stance times are a legitimate measure for mobility or remove this statement from the paper.

RESPOND: 

Thank you for your comment. We have removed the word “mobility” and revised the sentence. Please refer to lines 243-245 and table 2. 

Line 220: you state that the arch-support insole absorbed the shock associated with the foot contacting the ground. Again, this is a pure speculation and you have not provided any evidence that this is the case. Please provide the evidence or delete this statement from the manuscript.

RESPOND: 

Thank you for your comment. We delete this statement from the manuscript. Please refer to line 246. 

Lines 238-239: I do not understand this sentence. Please elaborate what you want to say here. 

RESPOND: 

Thank you for your comment. We have revised the sentence. Please refer to lines 262-263. 

Lines 244-245: It is only a speculation that wearing arch-support insoles will “restore a normal, elastic arch in people with flatfoot, …”. Please clearly state that this is only a speculation or include a reference that has shown this in the past. Also, what is a “normal” arch? 

RESPOND: 

Thank you for your comment. We have revised the sentence and included a reference. Please refer to lines 267-269. 

Lines 247-249: You cannot make this statement. You have not demonstrated why reducing peak pressures would be a benefit for people with flatfoot. Please make a better case of how you link peak pressure to substantial benefits in people or remove this statement.

RESPOND: 

Thank you for your comment. We have revised the sentence. Please refer to lines 269-272. 

Lines 256-258: How does wearing an arch-support insole restore a “normal”, elastic arch? What do you speculate is the mechanism behind it? Again, what is “normal”? Perhaps, you should not use this term.

RESPOND: 

Thank you for your comment. We have revised the sentence. Please refer to lines 281-283. 

Lines 258-265: You have not assessed arch deformation in any way in this study; yet, you discuss the windlass mechanism and how the arch may or may not collapse with these insoles. I would strongly recommend highlighting that these are speculations, only, or remove the entire paragraph from the manuscript.

RESPOND: 

Thank you for your comment. We have removed the entire paragraph from the manuscript. Please refer to lines 284-286. 

Lines 269: how would the forefoot play a crucial role in the braking during downhill walking if all participants were striking with the heel? The forefoot will mainly still be in the air during the braking period of the stance phase. Therefore, the forefoot cannot contribute to the braking at all. Again, this is a speculation because you have not assessed forefoot motion. Please remove this statement from the paper.

RESPOND: 

Thank you for your comment. We have removed this statement from the paper. Please refer to line 290. 

Lines 276-278: If the area of the midfoot increases relative to the total area, then the foot is flatter compared to when the midfoot area is lower compared to the total area. This is also called the arch index (Cavanagh, 1987). Your conclusion, therefore, does not make sense. Please revise.

RESPOND: 

Thank you for your comment. In the current study, the contact area of the midfoot was measured with pressure insoles with sensors that were placed between the tested insole and foot during walking. The method of the measurement of the contact area for dynamic walking movement is kind of different from that of Cavanagh’s (1987) arch index for a static standing posture that usually uses a scanner with bare foot on it. The measurement of Cavanagh’s (1987) arch index is usually used to determine a flatter foot in a static standing posture. Nevertheless, the current study used Cavanagh’s formula to determine changes of the distribution of the parts of the foot when it was supported by the arch-support insole in walking movement. Using the arch-support insole like using foot arch-support orthotics inserted beneath the midfoot makes the contact area of the midfoot increased during walking. We have elaborated this statement in the method and this paragraph. Please refer to lines 288-293 and 168-172. 

Line 281: remove “make” from the sentence.

RESPOND: 

Thank you for your comment. We have removed “make” from the sentence. Please refer to line 291. 

Lines 293-294: you cannot make the statement that wearing the arch-support insole provided many benefits in mobility. You have not provided clear reasoning as to why this would be the case. Please remove this statement from the paper.

RESPOND: 

Thank you for your comment. We removed this statement from the paper. Please refer to lines 306-30. 

RESPOND to Reviewer #2:

1.The statistics need to change. There was a comparison of insole condition across different levels of incline. The results seemed to indicate that the variables that were different across insole conditions seemed to be dependent on the level of incline. With this observation in mind, the researchers should re-run the statistics on each dependent variable and instead of conducting multiple uncorrected t-tests, they should conduct a two way repeated measures ANOVA with factors INCLINE (uphill, level, downhill) and INSOLE (flat, arch support). This will indicate if there is an interaction effect for the measured pressure and gait variables.

RESPOND:

Thank you for your comment. We considered the reviewer’s comment to conduct a two-way repeated measures ANOVA with factors. The purpose of this study was mainly to compare the differences between the arch-support insole and flat insole in each of three slopes, respectively. Therefore, some of significance of the slopes did not discussed in the manuscript. Please refer to lines 178, 189-240 of the methods section, results and tables 2-4.

2. Why did you restrict you sample to only females, particularly when the condition seems to be more common in males? Please provide an explanation why there wasn’t any males.

RESPOND: 

Thank you for your comment. We currently only recruit females this time in case of some homogeneity issues. We will further recruit male participants and compare the difference between males and females. 

3. Please provide an explanation as to why your static arch index was representative of flatfoot. My understanding is that flat foot would have something to do with the proportion of the foot that is or is not in contact with the ground in a loaded or unloaded condition. What does the narrowest and widest part of the foot have to do with the arch collapsing or per planus?

RESPOND:

Thank you for your comment. There are around 30 methods to evaluate the flatfoot (Banwell et. al, 2018; Cavanagh & Rodgers, 1987). There are two conditions of measuring the static arch index. One is in a body-weight loaded condition; another is in an body-weight unloaded condition. The current study used the measurement in a body-weight loaded condition. The static arch index is related to the ratio between the smallest width of the midfoot and the largest width of the metatarsal head area (Gill, Lewis, and DeSilva, 2014). This is also called the Chippaux and Smirakarc index (CSI) developed by Chippaux and Smirakarch. The CSI is correlated with skeletal measures of arch height such as the navicular height. People with flatfoot tend to have lower arch height during a stance based on a footprint. Then, the narrowest part of the foot will increase and change the CSI. The CSI had reliability and validity to measure and define flat foot populations (Gill, Lewis, and DeSilva, 2014).

Reference:

Banwell HA, Paris ME, Mackintosh S, Williams CM. Paediatric flexible flat foot: how are we measuring it and are we getting it right? A systematic review. Journal of Foot & Ankle Research. 2018; 11(1): 21-33. doi:10.1186/s13047-018-0264-3

Cavanagh PR, Rodgers MM. The arch index: a useful measure from footprints. Journal of Biomechanics. 1987; 20(5): 547–51. doi: 10.1016/0021-9290(87)90255-7 

Gill SV, Lewis CL, DeSilva JM. Arch Height Mediation of Obesity-Related Walking in Adults: Contributors to Physical Activity Limitations. Physiology Journal. 2014; Article ID 821482 doi: 10.1155/2014/821482

4. Where was the pressure insole placed relative to the insoles (i.e., underneath or above)? Please provide an explanation why you used on placement method over the other.

RESPOND:

Thank you for your comment. Wireless pressure insole with the F-Scan sensor (Model #3000E Tekscan, Inc., Boston, MA, USA) were placed above either the tested flat insole or arch-support insole of each foot to detect participant’s plantar pressure. The pressure insole was placed between the tested insole and foot to measure the contact pressure variables of the foot on the tested insole. If the pressure insole is placed underneath the tested insole, the contact pressure variables would be the tested insole rather than the foot and they may be affected by many other factors such as the thickness and the shape of the insole itself. 

5. How much of the results were due to hardness/stiffness of the insole versus it having arch support or not? For example, there was a difference in midsole hardness in the arch support condition and this condition also presented with an increase contact area. The same results applies to the forefoot area such that the harder condition showed increase contact area.

RESPOND:

Thank you for your comment. The arch-support insole provide support in the midsole with greater hardness compared with the flat insole. However, the current study cannot show how much of the results were due to hardness of the insole versus it having arch support or not. It has been claimed in the limitation. Please refer to lines 301-303.

6. Please change “gait process time” to stance time that would be more consistent with the literature.

RESPOND:

Thanks for the comment. We have revised this throughout the manuscript. Please refer to lines 48, 54, 67, 107, 161, 163, 189, 190, 192, 193, 194, 250, 252, 254 and 258.

7. Please present the data across the dominant and non-dominant leg. If in one leg there was higher peak pressure and lower peak pressure in the other, this would average out to no change in pressure. I don’t think your averaging method avoids discrepancies.

Please incorporate these changes and adjust the discussion accordingly. Once those comments are addressed, I will be able to provide more feedback on the minor, or major, comments that remain. 

RESPOND:

Thanks for the comment. We provided the data of dominant and non-dominant leg. Please refer to table 2-4. In addition, foot pressure was collected by F-Scan sensor (Model #3000E Tekscan, Inc., Boston, MA, USA) in the stance phase for each foot. In a stance phase, the F-Scan sensor counted the support foot pressure, while the swing foot pressure did not be counted. We averaged data of dominant and non-dominant leg measured during stance phases.

---

## [Decision Letter · Decision Letter 1]

3 Jun 2020

PONE-D-20-05316R1

The arch support insoles show benefits to people with flatfoot on stance time, plantar pressure and contact area

PLOS ONE

Dear Dr. Chen-Yi Song,

Thank you for submitting your manuscript to PLOS ONE. After careful consideration, we feel that it has merit but does not fully meet PLOS ONE’s publication criteria as it currently stands. Therefore, we invite you to submit a revised version of the manuscript that addresses the points raised during the review process.

We look forward to receiving your revised manuscript.

Kind regards,

Gianluca Vernillo, Ph.D.

Academic Editor

PLOS ONE

Reviewers' comments:

Reviewer's Responses to Questions

**Comments to the Author**

1. If the authors have adequately addressed your comments raised in a previous round of review and you feel that this manuscript is now acceptable for publication, you may indicate that here to bypass the “Comments to the Author” section, enter your conflict of interest statement in the “Confidential to Editor” section, and submit your "Accept" recommendation.

Reviewer #1: (No Response)

Reviewer #2: (No Response)

2. Is the manuscript technically sound, and do the data support the conclusions?

Reviewer #1: Yes

Reviewer #2: Yes

3. Has the statistical analysis been performed appropriately and rigorously? 

Reviewer #1: Yes

Reviewer #2: No

4. Have the authors made all data underlying the findings in their manuscript fully available?

Reviewer #1: Yes

Reviewer #2: Yes

5. Is the manuscript presented in an intelligible fashion and written in standard English?

Reviewer #1: Yes

Reviewer #2: Yes

6. Review Comments to the Author

Reviewer #1: I have only one question left to the authors addressing my first comment of the first round of revisions:

I think that the authors made a great attempt to link their variables of interest to measures of importance to people with flatfeet. However, the authors attempted to link simple plantar pressure variables to mobility and walking benefits without describing what is meant by these terms. I would recommend to clearly state what is meant by the terms "mobility" and "benefits" and how improvements in mobility look like.

Authors' response:

Thank you for your comment. We have changed the term "mobility" to gait speed. Studenski, et al. (2011) indicated that longer stance time reflects lower gait speed that is predictive of increased mortality in middle-aged and elderly people. The shortened stance time found in the current study was speculated to increase the gait speed because of wearing the arch-support insole. In addition, we have removed this statement from conclusion that "wearing the arch-support insole provided many benefits in mobility". Please refer to lines 242-247. These results imply that wearing an arch-support insole provides benefits in the shortened stance time and generation of propulsion force to the big toe while walking on uphill and level surfaces and to the metatarsals 2-4while walking on the level surface. We have stated these in abstract and conclusion section. Please refer to lines 61-63 and 306-310.

Reference:

Studenski S, Perera S, Patel K, Rosano C, Faulkner K, Inzitari M, et al. Gait speed and survival in older adults. Jama. 2011; 305(1): 50-58. doi:10.1001/jama.2010.1923

New question:

How can gait speed be increased if your participants were walking at a set speed on a treadmill? According to your methods in line 141, your participants were walking at 0.75 m/s. I would hope that their gait speed did not change between conditions. The stance times may have changed but this would infer a change in step frequency or swing time as well. Was this the case?

Reviewer #2: Thank you to the authors for making some of the requested changed. The authors will still need to address the following comments:

1. Please provide the rationale as to why only females were recruited in the text of the manuscript under the methods section.

2. You did not provide a sufficient statement of the hardness of the insoles as a limitation. You need to be specific. You should be stating that the arch support insole did have higher hardness in the midsole and the flat insole was harder in the forefoot and the heel. This difference in stiffness could be a confounding variable to the results in the study and therefore, you cannot conclude that the differences were due to differences in hardness or the presence vs absence of an arch support

3. Your purpose statement reads “the purpose of this study was to investigate the effects of the arch-support insoles on stance time, cadence, peak pressure, and contact area of the foot with the ground while walking uphill, downhill, and on a level surface, respectively.” In the discussion, you did not provide much explanation of the interaction effects that were found. You need to perform a post-hoc analysis on the interaction effects to determine the location of the interaction effect. For example, stance time had an interaction effect across insole and slope. It appears that this difference occurred in level walking, but not uphill and downhill walking. This needs to be determined with the post-hoc analysis and then an interpretation should be provided in the discussion. Please complete this type of analysis and interpretation for each interaction effect.

7. PLOS authors have the option to publish the peer review history of their article (what does this mean?). If published, this will include your full peer review and any attached files.

Reviewer #1: Yes: Sasa Cigoja

Reviewer #2: No

---

## [Author Response · Author response to Decision Letter 1]

7 Jul 2020

Reviewer #1: I have only one question left to the authors addressing my first comment of the first round of revisions:

I think that the authors made a great attempt to link their variables of interest to measures of importance to people with flatfeet. However, the authors attempted to link simple plantar pressure variables to mobility and walking benefits without describing what is meant by these terms. I would recommend to clearly state what is meant by the terms "mobility" and "benefits" and how improvements in mobility look like.

Response:

Thank you for your comment. We have removed the term "mobility" throughout the discussion. We focused to mention the benefits to the people with flatfeet from wearing the arch-support insole in the shortened stance time when walking on a level treadmill found in the current study. It was speculated to increase the gait speed when walking on the level ground. Studenski, et al. (2011) indicated that longer stance time reflects lower gait speed that is predictive of increased mortality in middle-aged and elderly people. We have stated these in the abstract and conclusion sections. Please refer to lines 71-73, 277-279 and 337-340.

Reference:

1. Studenski S, Perera S, Patel K, Rosano C, Faulkner K, Inzitari M, et al. Gait speed and survival in older adults. Jama. 2011; 305(1): 50-58. doi:10.1001/jama.2010.1923

New question:

How can gait speed be increased if your participants were walking at a set speed on a treadmill? According to your methods in line 141, your participants were walking at 0.75 m/s. I would hope that their gait speed did not change between conditions. The stance times may have changed but this would infer a change in step frequency or swing time as well. Was this the case?

RESPOND:

Thank you for your comment. We agreed with the review’s comment that the gait speed would not change when walking at a set speed on a treadmill. We have rephrased the sentence and further investigated changes in step frequency. We found that there was an interaction between the insole and slope in the step frequency. In addition, the interaction with marginal significance between insoles and slopes was found in the step frequency (p=0.044). Simple main effects of insoles showed that the step frequency of the uphill and level surfaces was significantly shorter than that of the downhill surface in the arch-support (p< 0.001, ES=1.22; p< 0.001, ES=2.05, respectively) and flat (p=0.004, ES=1.02; p< 0.001, ES=1.75, respectively) insole. Simple main effects of slopes showed that the step frequency has no difference between the insoles on the uphill (p=0.132, ES=0.10), downhill (p=0.346, ES=0.10), and level (p=0.053, ES=0.28) surface. Please refer to lines 214-221 and Table 2.

Reviewer #2: Thank you to the authors for making some of the requested changed. The authors will still need to address the following comments:

1. Please provide the rationale as to why only females were recruited in the text of the manuscript under the methods section.

RESPOND:

Thank you for your comment. Previous study has estimated the prevalence of mild and severe cases of the flatfoot to be 16.2% among males and 11.7% among females which was close to each other in gender (Tenenbaum et al., 2013). Nevertheless, females are more likely to suffer from risks of lower extremity injuries in running (Alomonroeder & Benson, 2017) and lack of flexible foot/shank coupling coordination compared to males (Noghondar & Yazdi, 2017). We have provided the rational in the introduction. Please refer to lines 83-86 .

References:

(1) Tenenbaum S, Hershkovich O, Gordon B, Bruck N, Thein R, Derazne E, et al. Flexible pes planus in adolescents: body mass index, body height, and gender--an epidemiological study. Foot & Ankle International. 2013; 34(6): 811-817. doi: 10.1177/1071100712472327

(2) Alomonroeder TG, Benson LC. Sex differences in lower extremity kinematics and patellofemoral kinetics during running. J Sport Sci. 2017; 35(16): 1575-1581. doi: 10.1080/02640414.2016.1225972 

(3) Noghondar FA, Yazdi NK. Assessment of patterns and variability in lower extremity coordination between genders with different shoe insole stiffness during jump-landing tasks. Human Movement. 2017; 18(1): 37–43. doi: 10.1515/humo-2017-0002

2. You did not provide a sufficient statement of the hardness of the insoles as a limitation. You need to be specific. You should be stating that the arch support insole did have higher hardness in the midsole and the flat insole was harder in the forefoot and the heel. This difference in stiffness could be a confounding variable to the results in the study and therefore, you cannot conclude that the differences were due to differences in hardness or the presence vs absence of an arch support

RESPOND:

Thank you for your comment. We have added the review’s comment regarding the hardness of the insoles in the limitation section. Please refer to lines 330-334.

3. Your purpose statement reads “the purpose of this study was to investigate the effects of the arch-support insoles on stance time, cadence, peak pressure, and contact area of the foot with the ground while walking uphill, downhill, and on a level surface, respectively.” In the discussion, you did not provide much explanation of the interaction effects that were found. You need to perform a post-hoc analysis on the interaction effects to determine the location of the interaction effect. For example, stance time had an interaction effect across insole and slope. It appears that this difference occurred in level walking, but not uphill and downhill walking. This needs to be determined with the post-hoc analysis and then an interpretation should be provided in the discussion. Please complete this type of analysis and interpretation for each interaction effect.

RESPOND:

Thank you for your comment. We have performed a post-hoc analysis on the interaction effects and provided information regarding the interaction effects in the discussion section. Please refer to lines 273-275, 289-291, 301-303, and 305-307.

---

## [Decision Letter · Decision Letter 2]

16 Jul 2020

PONE-D-20-05316R2

The arch support insoles show benefits to people with flatfoot on stance time, plantar pressure and contact area

PLOS ONE

Dear Dr. Song,

Thank you for submitting your manuscript to PLOS ONE. After careful consideration, we feel that it has merit but does not fully meet PLOS ONE’s publication criteria as it currently stands. Therefore, we invite you to submit a revised version of the manuscript that addresses the points raised during the review process.

Please make sure to properly address the reviewer's #2 comment. Your revised manuscript must not be open to the criticism that potentially could add confusion. Addressing and properly characterizing your sample is an essential prerequisite. Furthermore, according to the PLOS ONE's policy and the reviewer's #1 comment, please make your data publicly available.

We look forward to receiving your revised manuscript.

Kind regards,

Gianluca Vernillo, Ph.D.

Academic Editor

PLOS ONE

Reviewers' comments:

Reviewer's Responses to Questions

**Comments to the Author**

1. If the authors have adequately addressed your comments raised in a previous round of review and you feel that this manuscript is now acceptable for publication, you may indicate that here to bypass the “Comments to the Author” section, enter your conflict of interest statement in the “Confidential to Editor” section, and submit your "Accept" recommendation.

Reviewer #1: All comments have been addressed

Reviewer #2: (No Response)

2. Is the manuscript technically sound, and do the data support the conclusions?

Reviewer #1: Yes

Reviewer #2: Yes

3. Has the statistical analysis been performed appropriately and rigorously? 

Reviewer #1: Yes

Reviewer #2: Yes

4. Have the authors made all data underlying the findings in their manuscript fully available?

Reviewer #1: No

Reviewer #2: Yes

5. Is the manuscript presented in an intelligible fashion and written in standard English?

Reviewer #1: Yes

Reviewer #2: Yes

6. Review Comments to the Author

Reviewer #1: Thank you for addressing my comments. I would recommend that the paper is accepted for publication once the set of data that is needed to reproduce the findings and conclusions of this study are made publicly available (e.g., online data repository) as per PLOS Data policy.

Reviewer #2: You have addressed two of my comments thoroughly and one partially. For comment 1 with respect to recruiting only females, you need to state this reason explicitly in the methods section after you mention that only females were recruited. In its current form, this selection criteria is unclear. Please insert a comment at the beginning of the methods section that addresses this choice.

7. PLOS authors have the option to publish the peer review history of their article (what does this mean?). If published, this will include your full peer review and any attached files.

Reviewer #1: No

Reviewer #2: No

---

## [Author Response · Author response to Decision Letter 2]

22 Jul 2020

Reviewer #1: Thank you for addressing my comments. I would recommend that the paper is accepted for publication once the set of data that is needed to reproduce the findings and conclusions of this study are made publicly available (e.g., online data repository) as per PLOS Data policy.

Response:

Thank you for your comment. We uploaded the data to the online data repository as per PLOS Data policy. 

Reviewer #2: You have addressed two of my comments thoroughly and one partially. For comment 1 with respect to recruiting only females, you need to state this reason explicitly in the methods section after you mention that only females were recruited. In its current form, this selection criteria is unclear. Please insert a comment at the beginning of the methods section that addresses this choice.

Response:

Thank you for your comment. We have inserted a comment at the beginning of the methods section that addresses this choice. Please refer to lines 124-126.

---

## [Decision Letter · Decision Letter 3]

27 Jul 2020

The arch support insoles show benefits to people with flatfoot on stance time, plantar pressure and contact area

PONE-D-20-05316R3

Dear Dr. Song,

We’re pleased to inform you that your manuscript has been judged scientifically suitable for publication and will be formally accepted for publication once it meets all outstanding technical requirements.

Kind regards,

Gianluca Vernillo, Ph.D.

Academic Editor

PLOS ONE

Additional Editor Comments (optional):

Reviewers' comments:

Reviewer's Responses to Questions

**Comments to the Author**

1. If the authors have adequately addressed your comments raised in a previous round of review and you feel that this manuscript is now acceptable for publication, you may indicate that here to bypass the “Comments to the Author” section, enter your conflict of interest statement in the “Confidential to Editor” section, and submit your "Accept" recommendation.

Reviewer #2: All comments have been addressed

2. Is the manuscript technically sound, and do the data support the conclusions?

Reviewer #2: Yes

3. Has the statistical analysis been performed appropriately and rigorously? 

Reviewer #2: Yes

4. Have the authors made all data underlying the findings in their manuscript fully available?

Reviewer #2: Yes

5. Is the manuscript presented in an intelligible fashion and written in standard English?

Reviewer #2: Yes

6. Review Comments to the Author

Reviewer #2: (No Response)

7. PLOS authors have the option to publish the peer review history of their article (what does this mean?). If published, this will include your full peer review and any attached files.

Reviewer #2: No

---

## [Editor Report · Acceptance letter]

3 Aug 2020

PONE-D-20-05316R3 

The arch support insoles show benefits to people with flatfoot on stance time, plantar pressure and contact area 

Dear Dr. Song:

I'm pleased to inform you that your manuscript has been deemed suitable for publication in PLOS ONE. Congratulations! Your manuscript is now with our production department. 

Kind regards, 

on behalf of

Dr. Gianluca Vernillo 

Academic Editor

PLOS ONE